# Exosomes Derived from Bone Marrow Mesenchymal Stem Cells Promote Angiogenesis in Ischemic Stroke Mice via Upregulation of MiR-21-5p

**DOI:** 10.3390/biom12070883

**Published:** 2022-06-24

**Authors:** Hui Hu, Xiaowei Hu, Lin Li, Yan Fang, Yan Yang, Jingjing Gu, Jiadong Xu, Lisheng Chu

**Affiliations:** 1Department of Physiology, Zhejiang Chinese Medical University, Hangzhou 310053, China; huhui190756@cqmu.edu.cn (H.H.); 20181015@zcmu.edu.cn (X.H.); lilin@zcmu.edu.cn (L.L.); fangyan@zcmu.edu.cn (Y.F.); yangyan@zcmu.edu.cn (Y.Y.); 20201007@zcmu.edu.cn (J.X.); 2Department of Pathology, Zhejiang Chinese Medical University, Hangzhou 310053, China; 20141030@zcmu.edu.cn

**Keywords:** bone marrow mesenchymal stem cells, exosome, microRNA, angiogenesis, cerebral ischemia

## Abstract

Exosomes derived from bone mesenchymal stem cells (BMSC-Exos) are one of the main factors responsible for the therapeutic effects of BMSCs. The study aimed to investigate whether BMSC-Exos could promote angiogenesis in ischemic stroke mice via miR-21-5p. In ischemic stroke mice, the therapeutic effects of BMSC-Exos were evaluated by neurological functions and infarct volume. Microvessel density was detected by BrdU/vWF immunofluorescence staining. In in vitro experiments, the proangiogenic effects of BMSC-Exos were assessed via proliferation, migration, and tube formation of human umbilical vein endothelial cells (HUVECs). The miR-21-5p inhibitor was transfected into BMSCs using Lipofectamine 2000. miR-21-5p expression was detected by qRT-PCR. The expression levels of VEGF, VEGFR2, Ang-1, and Tie-2 were determined by Western blot. BMSC-Exos significantly improved neurological functions and reduced infarct volume, upregulated microvessel density, and miR-21-5p expression after cerebral ischemia. In vitro assays revealed that BMSC-Exos enhanced HUVECs functions including proliferation, migration, and tube formation. BMSC-Exos increased the expression levels of VEGF, VEGFR2, Ang-1, and Tie-2. However, the proangiogenic effects of BMSC-Exos on HUVECs were reversed by the miR-21-5p inhibitor. These results suggest that BMSC-Exos could promote angiogenesis via miR-21-5p upregulation, making them an attractive treatment strategy for stroke recovery.

## 1. Introduction

Stroke is one of the leading causes of death and disability worldwide [1]. Up to now, the only recommended drug for the treatment of acute cerebral ischemia is tissue-type plasminogen activator (tPA), which works as a thrombolytic agent. However, the narrow therapeutic time window and potentially fatal hemorrhage severely limit its clinical application [2]. Therefore, novel effective strategies designed to improve functional recovery after ischemic stroke are urgently needed.

A growing number of studies have indicated that stem cell-based therapy represents a new approach to the treatment of ischemic stroke [3]. Mesenchymal stem cells (MSCs) can be obtained easily and expanded rapidly in vitro, which makes them an ideal candidate for cell-based therapy [4]. A variety of studies have demonstrated that transplanted MSCs could promote neurogenesis and angiogenesis, thus facilitating stroke recovery [5,6]. As for the underlying mechanism of these functions, it was initially believed that MSCs could be home to ischemic tissues and differentiated to replace injured cells [7]. However, subsequent studies reported poor survival and engraftment of transplanted MSCs in ischemic regions, which was not sufficient to explain the mechanism of MSCs differentiation [8]. Until now, it is proposed that MSCs exert their beneficial effects mainly by secreting paracrine factors, such as trophic factors and extracellular vesicles (EVs) [9,10].

Exosomes (Exos) are small EVs sized approximately 30–150 nm in diameter. They are secreted by a variety of cell types and contain biological molecules, including proteins, lipids, mRNAs, and microRNAs (miRs) [11]. The beneficial effects of MSCs-derived Exos on stroke have been particularly studied, but the mechanisms of action are still not fully clarified [12,13]. Recent studies have shown that Exos transfer miRs to the recipient cells in the brain to affect nervous and vascular systems and ultimately alleviate cerebral ischemic injury [14,15,16].

miRs are small, single-stranded non-coding RNAs that bind to target mRNAs and suppress protein expression by inhibiting mRNAs translation or degrading [17]. They are engaged in many physiological and pathological processes such as cell proliferation, migration, differentiation, metabolism, and apoptosis [18]. Increasing evidence has demonstrated that miRs are important regulators of angiogenesis. MiR-21-5p notably promoted local neovascularization in myocardial infarction model [19]. Exos derived from endothelial progenitor cells could promote endothelial cell repair by transferring miR-21-5p [20]. Recently, the miRNA expression profile revealed that miR-21-5p was abundant in MSC-Exos [21]. Exosomes derived from BMSCs with the stimulation of Fe_3_O_4_ nanoparticles and a static magnetic field could enhance wound healing through upregulated miR-21-5p [22]. However, it is not clear whether BMSC-Exos promote angiogenesis after cerebral ischemia through miR-21-5p upregulation.

In this study, we firstly studied the putatively beneficial effects of BMSC-Exos on neurological function and angiogenesis in cerebral ischemia and then confirmed whether these proangiogenic effects were related to miR-21-5p transferred by BMSC-Exos in vitro.

## 2. Materials and Methods

### 2.1. Animals

Male Sprague-Dawley rats weighing 80–100 g (3 weeks old) and male ICR mice weighing 25–30 g (8–10 weeks old) were obtained from SIPPR/BK Laboratory Animal (Shanghai, China). The animals were kept at stable temperature (22 ± 2 °C) and humidity (60 ± 5%) conditions with 12-h light/dark cycle and free access to water and food. All animal care and experimental procedures were approved by the Experimental Animal Care and Ethics Committee of Zhejiang Chinese Medical University (reference number: ZSLL-2017-058) and were performed in accordance with the National Institutes of Health Guidelines for the Care and Use of Laboratory Animals.

### 2.2. Isolation and Culture of BMSCs

BMSCs were isolated based on our previously described method [23]. Briefly, bone marrow was flushed with prechilled DMEM/F12 containing 1% (*v*/*v*) penicillin and streptomycin (Haotian Biological Technology, Hangzhou, China) from rat femurs and tibias. Bone marrow aspirates were then centrifuged and the cells suspended in DMEM/F12 supplemented with 10% fetal bovine serum (FBS) (Gibco, Thermo Fisher Scientific, Waltham, MA, USA). The medium was replaced 2 days after and every 3 days thereafter. The cells were passaged when 80–90% confluence was reached with a ratio of 1:2. Passage 3–4 BMSCs were used for the following experiments.

### 2.3. Characterization of BMSCs

BMSCs phenotype was determined by flow cytometry using specific antibodies against cell surface markers, including CD34 (Santa Cruz, Santa Cruz, CA, USA), CD29, CD45, and CD90 (Biolegend, San Diego, CA, USA). To evaluate osteogenic and adipogenic differentiation capacity of BMSCs, cells were cultured in 6-well plates at a density of 2 × 10^4^ cells/cm^2^ with osteogenic and adipogenic differentiation medium (Cyagen, Guangzhou, China) according to manufacturer’s protocol. After 2 weeks, cells were fixed in 4% paraformaldehyde, and then stained with Alizarin Red S or Oil Red O, respectively.

### 2.4. Culture of Human Umbilical Vein Endothelial Cells (HUVECs)

HUVECs obtained from American Type Culture Collection (ATCC, Manassas, VA, USA) were cultured in RPMI-1640 medium (Hyclone, Logan, UT, USA) containing 10% FBS and 1% antibiotics. and were maintained at 37 °C with 5% CO_2_. HUVECs were passaged every two or three days.

### 2.5. BMSC-Exos Isolation, Purification, and Identification

BMSC-Exos were isolated and purified by differential ultracentrifugation method [24]. Briefly, upon reaching 70–80% confluence, BMSCs were rinsed three times with PBS and cultured in fresh medium containing 10% Exos-free FBS medium (SBI Biosciences, Palo Alto, CA, USA). The supernatants were collected after an additional 48 h incubation and sequentially centrifuged at 300× *g* and 2000× *g* for 10 min to remove dead cells. Then, the supernatants were centrifuged at 10,000× *g* for 30 min at 4 °C to remove residual cellular debris and filtered with a 0.22-μm filter (Millipore, Billerica, MA, USA). Afterward, the filtrates were ultracentrifuged at 4 °C and 100,000× *g* for 2 h, followed by washing with PBS and ultracentrifuged at 100,000× *g* for 2 h. At last, the pelleted Exos were resuspended in 100 μL PBS and total proteins were quantified using a Micro BCA Protein Assay kit (Thermo Scientific, Rockford, IL, USA), then stored at −80 °C.

For identification of Exos, the morphology, particle concentration, size distribution, and specific surface markers (CD9, CD63, and TSG101) of isolated Exos were detected by transmission electron microscope (TEM, Hitachi, Japan), nanoparticle tracking analysis (NTA), and Western blot analysis, respectively.

### 2.6. Focal Cerebral Ischemia Model in Mice

Focal cerebral ischemia was induced by middle cerebral artery occlusion (MCAO) as described previously [25]. Briefly, mice were anesthetized with 2% isoflurane, and right common carotid artery (CCA), external carotid artery (ECA), and internal carotid artery (ICA) were carefully isolated. Then, a tip blunted and poly-L-lysine coated 6-0 nylon monofilament suture was inserted through right ECA and was advanced approximately 10 mm distal to the right carotid bifurcation to occlude the origin of MCA. After 60 min of ischemia, the suture was carefully withdrawn. Sham-operated mice received identical surgery without suture insertion. During the surgery, rectal temperature of mice was maintained at 37 °C with a homeothermic pad.

### 2.7. BMSC-Exos Administration and BrdU Labeling

To evaluate therapeutic effects of BMSC-Exos, mice were divided into four cohorts: sham operation group, MCAO group, MCAO plus 25 μg BMSC-Exos group (25 μg BMSC-Exos group), and MCAO plus 50 μg BMSC-Exos group (50 μg BMSC-Exos group), and then Exos were administered in 100 μL PBS or PBS alone via tail vein injection at 24 h after ischemia. To observe cell proliferation, 5-bromo-2-deoxyuridine (BrdU, Sigma-Aldrich, St. Louis, MO, USA) of 50 mg/kg was administered in mice via intraperitoneal injection 24 h after ischemia, followed by daily consecutive injection for 14 days.

### 2.8. Neurological Function Evaluation

In all animals, behavioral tests were performed before MCAO and at 1, 3, 7, and 14 days after MCAO by an investigator who was blinded to experimental group design. Neurological deficit score was evaluated according to Zea Longa score [26]: 0, no deficit; 1, failure to fully extend left forepaw; 2, circling to the left; 3, paresis to the left; 4, depressed level of consciousness and no spontaneous walking. Corner test was carried out as described [27]. Briefly, two 30 cm × 20 cm × 1 cm boards were attached to each other at an angle of 30° and with a small opening between the two boards. A mouse was placed into the central square facing the corner. When both sides of the vibrissae were stimulated by boards, the mouse then reared forward and upward, after which it turned back to face the open end. Each mouse was tested for ten trials, and the selected turning sides were recorded.

### 2.9. Infarct Volume Assessment

Mice were sacrificed 3 and 14 days after MCAO, and brains were removed and frozen immediately at −20 °C for approximately 5 min and then dissected into 1 mm-thick coronal slices. The slices were stained with 2% 2,3,5-triphenyltetrazolium chloride (TTC) for 15 min and fixed with 4% paraformaldehyde for 24 h. Infarct volume was evaluated by Image J software, and infarct volume percentage was calculated as follows: infarct volume (%) = [left hemisphere volume − (right hemisphere volume − infarct volume)]/left hemisphere volume × 100%.

### 2.10. Immunofluorescence Staining

Mice were transcardially perfused with normal saline, followed by 4% paraformaldehyde solution for 10 min at 14 days after MCAO. Brains were fixed overnight at 4 °C, soaked in 30% sucrose solution, frozen, and cut into 10-μm-thick sections (Leica, Wetzlar, Germany). BrdU/von Willebrand factor (vWF), VEGF/vWF, VEGFR2/vWF, Ang-1/vWF, and Tie-2/vWF were detected by double immunofluorescence staining as described in our previous study [23]. Cellular nuclei were stained with 4′,6-diamidino-2-phenylindole (DAPI, Zsbio, Beijing, China) at room temperature.

### 2.11. BMSC-Exos Uptake by HUVECs

To label Exos with green fluorescent dye, Exos were firstly resuspended with PKH67 dye (Sigma-Aldrich, Munich, Germany) and incubated for 5 min, then terminated by 2% bovine serum albumin. Next, Exos were ultracentrifuged at 100,000× *g* for 1 h to remove unbound dyes. HUVECs were incubated with PKH67-labelled Exos for 12 h, followed by 4% paraformaldehyde fixation for 15 min and DAPI stain for 5 min. The internalization of PKH67-labeled Exos by HUVECs was observed using a fluorescence microscope (Leica, Wetzlar, Germany).

### 2.12. MiR-21-5p Inhibitor Transfection

miR-21-5p inhibitor and negative control (NC) were synthesized by RiboBio (Guangzhou Ribobio, Guangzhou, China). The sequences of miR-21-5p inhibitor and NC were 5′-UCAACAUCAGUCUGAUAAGCUA-3′ and 5′-CAGUACUUUUGUGUAGUACAAA-3′, respectively. BMSCs at 80% confluence were transfected with 100 nM miR-21-5p inhibitor or NC, which were performed using Lipofectamine 2000 and Opti-MEM medium (Invitrogen, Carlsbad, CA, USA) according to manufacturer’s protocol. After 6 h, the transfection mixture was replaced by DMEM/F12 containing 10% exosome-free FBS. Conditioned medium of transfected cells was collected and centrifuged as described above.

### 2.13. MTT Assay

The proliferation of HUVECs was measured using 3-(4,5-dimethyl-2-thiazolyl)-2,5-diphenyl-2-H-tetrazolium bromide (MTT) assay (Beyotime, Shanghai, China). HUVECs plated on 96-well plates (1 × 10^4^ cells/well) were cultured alone or cocultured with Exos (25 or 50 μg/mL) for 12, 24, 36, and 48 h. Then, 10 μL of MTT tetrazolium salt solution (5 mg/mL) was added to each well. After incubation for another 4 h, formazan crystals were dissolved by 150 μL addition of dimethyl sulfoxide (Sigma, St. Louis, MO, USA). The optical density (OD) value of each well was measured at 490 nm by microplate reader (Tecan Austria GmbH, Grodig, Austria). Each group was replicated six times.

### 2.14. Scratch Wound Healing Assay

HUVECs were seeded into 6-well plates (5 × 10^5^ cells/well), followed by scratching with 200 μL pipette tip when reaching 90% confluence. Then, 2 mL of serum-free RPMI-1640 medium supplemented with Exos was added to each well. Images were captured at 0 h and 24 h after wounding. Migration rate (%) was calculated as following: migration rate (%) = (initial wound area (t = 0 h) – residual area (t = 24 h))/initial wound area (t = 0 h) × 100%. Each group was triplicated.

### 2.15. Transwell Migration Assay

Transwell migration assay was carried out using 24-well chambers (8 μm, Corning, Corning, NY, USA). RPMI-1640 of 500 μL containing 1% FBS was added into the lower chamber, and HUVECs (6 × 10^4^ cells/well) suspended in 100 μL FBS-free medium were seeded in the upper chamber with or without Exos. After 8 h of migration, nonmigratory cells were removed from the top of the insert membrane using humidified cotton swabs. The migrated cells at the bottom surface of membrane were fixed in 4% paraformaldehyde and stained with 0.1% crystal violet. The migrated cells were imaged and counted at 5 random fields. Each group was triplicated.

### 2.16. Tube Formation Assay

After thawed overnight at 4 °C, 50 μL per well matrigel matrix (Corning, Bedford, MA, USA) was added into precooled 96-well plates and incubated at 37 °C to polymerize for 30 min. Next, HUVECs (2 × 10^4^ cells/well) in FBS-free RPMI-1640 containing Exos were seeded onto matrigel-coated plates. Capillary-like tubular structures were captured after 6 h incubation. Total tube lengths from five random microscopic fields were calculated using Angiogenesis Analyzer Image J software. Each group was triplicated.

### 2.17. Quantitative Reverse Transcription-Polymerase Chain Reaction (qRT-PCR)

Total RNA from brain tissues of ischemic boundary zone or cells was extracted using miRNeasy Mini Kit (Qiagen, Hilden, Germany). cDNA was produced from the total RNA using Mir-XTM miRNA First-strand Synthesis Kit (TaKaRa, Dalian, China). Subsequently, the product from reverse transcription was amplified with the SYBR Premix Ex Taq Kit (TaKaRa, Dalian, China) on an iQ5 real-time PCR detection system (Bio-Rad, Hercules, CA, USA). Relative expression levels of miRNA were calculated by 2^−ΔΔCt^ and were normalized to U6. Each sample was repeated 3 times and at least three samples obtained from independent experiments were examined. All primers used in this study are listed in Table 1.

### 2.18. Western Blot Analysis

Total protein from brain tissues of ischemic boundary zone or cells was extracted with RIPA lysis buffer containing protease inhibitor PMSF (Beyotime, Shanghai, China), and quantified using BCA protein assay kit (Beyotime, Shanghai, China). Firstly, equal amounts of protein were separated by 10% sodium dodecyl sulfate-polyacrylamide gel (SDS-PAGE) and transferred to PVDF membranes (Millpore, Billerica, CA, USA). After being blocked with 5% skim milk in TBST, the membranes were incubated with following primary antibodies overnight at 4 °C: rabbit anti-CD9 (1:1000; Bioworld, Minneapolis, MN, USA), anti-CD63 (1:1000; Bioworld, Minneapolis, MN, USA), anti-TSG101 (1:1000; Abcam, Cambridge, MA, USA), anti-VEGF (1: 500; Santa Cruz, Santa Cruz, CA, USA), anti-VEGFR2 (1:500; Abcam, Cambridge, MA, USA), anti-Tie-2 (1:1000; Santa Cruz, Santa Cruz, CA, USA), mouse anti-GAPDH (1:1000; Santa Cruz, Santa Cruz, CA, USA), and goat anti-Ang-1 (1:1000; Santa Cruz, Santa Cruz, CA, USA). Then, the membranes were incubated with horseradish peroxidase-conjugated secondary antibodies of goat anti-rabbit IgG (Cell Signaling Technology, Danvers, MA, USA), goat anti-mouse IgG (Thermo Fisher Scientific, Waltham, MA, USA) or donkey anti-goat IgG (Santa Cruz, CA, USA) at room temperature for 1 h. Signals were visualized by enhanced chemiluminescence detection kit (Millpore, CA, USA). Relative expression levels were normalized to GAPDH.

### 2.19. Statistical Analysis

Data were analyzed by SPSS software (version 25.0, SPSS, Chicago, IL, USA) and presented as mean ± standard error of mean (SEM) unless indicated otherwise. Neurological deficit and corner test data were analyzed by nonparametric Kruskal–Wallis H test. All other data were analyzed by one-way analysis of variance (ANOVA) followed by the Student–Newman–Keuls post-hoc test. *p* < 0.05 was considered statistically significant.

## 3. Results

### 3.1. Characterization of BMSCs and BMSC-Exos

BMSCs (P3) exhibited a typical spindle-shaped morphology (Figure 1A). After differentiation induction, calcium deposits and intracytoplasmic lipid droplets appear that were stained by Alizarin Red S or Oil Red O, respectively (Figure 1B,C), indicating that BMSCs had the potential for osteogenic and adipogenic differentiation. To characterize BMSCs phenotype, BMSCs surface markers were analyzed by flow cytometry. The results showed that these cells were strongly positive for CD29 (99.83%) and CD90 (99.88%) (mesenchymal stem-cell specific markers), whereas they were negative for CD34 (1.44%) and CD45 (0.30%) (hematopoietic cell-specific markers) (Figure 1D).

As presented in Figure 2A, BMSC-Exos had a typical spherical or cup-shaped morphology. NTA revealed a bell-shaped curve of BMSC-Exos size distribution with a peak at approximately 108.4 nm, and the average concentration of the particles was 4.8 × 10^7^ particles/mL (Figure 2B). Additionally, isolated BMSC-Exos expressed exosomal markers CD9, CD63, and TSG101 without the expression of non-exosome marker protein GAPDH (Figure 2C). These characteristics indicated that the BMSC-derived particles were Exos.

### 3.2. BMSC-Exos Ameliorated Ischemic Brain Injury in Mice

To determine whether BMSC-Exos improved neurological function recovery, the neurological deficit score and the corner test were performed before MCAO and at 1, 3, 7, and 14 days after MCAO in mice (Figure 3A). The results demonstrated that BMSC-Exos decreased the neurological deficit score and right-turn number in a dose-related manner (Figure 3B,C). Compared with the MCAO group, BMSC-Exos significantly reduced infarction volume three days after ischemia. At 14 days after ischemia, apparent atrophy of the ischemic hemisphere was observed in all groups of mice, except the sham-operated group. The cerebral atrophy in the BMSC-Exos group was markedly reduced compared to the MCAO group. In addition, compared with the 25 μg BMSC-Exos group, the infarct volume was significantly decreased in the 50 μg BMSC-Exos group at 3 and 14 days, respectively (Figure 3D,E).

### 3.3. BMSC-Exos Promoted Angiogenesis in the Ischemic Boundary Zone

Proliferative microvessel density was estimated by BrdU/vWF immunofluorescent staining on day 14 after MCAO. The number of BrdU^+^/vWF^+^ cells was significantly increased in the MCAO and BMSC-Exos groups compared with the sham group. Furthermore, microvessel density in the BMSC-Exos group was significantly higher than that in the MCAO group (Figure 4A,B).

### 3.4. BMSC-Exos Increased VEGF/VEGFR2 and Ang-1/Tie-2 Protein Expressions after MCAO in Mice

To test whether BMSC-Exos promotes angiogenesis via upregulating the expression of VEGF, VEGFR2, Ang-1, and Tie2, the expression levels of these proteins were detected by western blot analysis. Compared with the sham group, the expression levels of VEGF, VEGFR2, Ang-1, and Tie-2 were increased in the MCAO and BMSC-Exos group. Furthermore, the expression levels of these proteins were increased in the BMSC-Exos group compared with the MCAO group (Figure 5A,B). Meanwhile, double-labeling immunofluorescence indicated that VEGF, VEGFR2, Ang-1, and Tie-2 were co-localized with brain endothelial cells in the ischemic boundary zone (Figure 5C).

### 3.5. BMSC-Exos Increased miR-21-5p Expression after MCAO in Mice

To study the changes in specific miRs expression levels after stroke, the expressions of several candidate miRs, including let-7i-5p, miR-21-5p, miR-22-3p, and miR-486, were detected by qRT-PCR in the ischemic boundary zone at day 14 after MCAO. The expression levels of these miRs were increased in the BMSC-Exos group compared with the MCAO group (Figure 6). In particular, miR-21-5p expression levels were increased approximately 22-fold (Figure 6).

### 3.6. HUVECs Uptake BMSC-Exos

To confirm that BMSC-Exos could be internalized into HUVECs, BMSC-Exos were labeled with PKH67 and then added to HUVECs for 12-h incubation. Fluorescence images showed that PKH67-labeled BMSC-Exos were located in the cytoplasm of HUVECs (Figure 7A). This result revealed that our purified BMSC-Exos could be taken up by HUVECs.

### 3.7. BMSC-Exos Promoted HUVECs Angiogenesis by Transferring miR-21-5p

Since endothelial cell proliferation, migration, and sprouting are critical in angiogenesis, the effects of BMSC-Exos on HUVECs proliferation, migration, and tube formation were studied. The results indicated that BMSC-Exos could enhance the proliferation, migration, and tube formation of HUVECs (Figure 7B–H). Therefore, 50 μg/mL BMSC-Exos was selected for the following experiments.

To clarify whether BMSC-Exos promote angiogenesis of HUVECs by upregulation of specific miRs, the expression levels of let-7i-5p, miR-21-5p, miR-22-3p, and miR-486 in HUVECs were analyzed. The results showed that miR-21-5p expression was significantly increased in BMSC-Exos treated HUVECs (Figure 8A), which was consistent with the results in vivo. Then, BMSCs were transfected with miR-21-5p inhibitor or NC. qRT-PCR showed that miR-21-5p expression was significantly downregulated in BMSCs and BMSC-Exos (Figure 8B,C). Treatment with Exos derived from BMSCs transfected with miR-21-5p inhibitor (miR-21-5p-Exos) also significantly decreased miR-21-5p expression in HUVECs (Figure 8D). These results indicated that BMSC-Exos increased miR-21-5p expression in HUVECs by transferring miR-21-5p.

To further determine whether BMSC-Exos enhance angiogenesis of HUVECs via transport of miR-21-5p, we directly transfected BMSCs with miR-21-5p inhibitor. As shown in Figure 9A–G, the effects of BMSC-Exos on HUVECs proliferation, migration, and tube formation were blocked by the miR-21-5p inhibitor. Thus, our results indicated that the proangiogenic activity of BMSC-Exos was partially mediated by transferring miR-21-5p.

### 3.8. BMSC-Derived Exosomal miR-21-5p Enhanced VEGF and VEGFR2 Expressions in HUVECs

To examine the mechanisms of proangiogenic action of BMSC-Exos, BMSCs were transfected with miR-21-5p inhibitor, and VEGF and VEGFR2 protein expressions in HUVECs were detected by western blot analysis. Results showed that BMSC-Exos remarkably increased the expression levels of VEGF and VEGFR2. However, this effect was reversed by a miR-21-5p inhibitor (Figure 9H–J). The above results revealed that BMSC-Exos enhanced VEGF and VEGFR2 expressions in HUVECs by transferring miR-21-5p.

## 4. Discussion

In the present study, we firstly confirmed that systemic treatment of BMSC-Exos could improve cerebral ischemia injury and promote angiogenesis in mice. Simultaneously, BMSC-Exos increased proangiogenic protein expression, including VEGF, VEGFR2, Ang-1, and Tie-2, and upregulated miR-21-5p expression in ischemic boundary regions. Then, the miR-21-5p inhibitor was transfected into BMSCs, which further confirmed that BMSC-Exos improved HUVECs angiogenesis via miR-21-5p transfer in vitro.

Increasing evidence suggests that BMSCs exert their therapeutic effects through paracrine mechanisms, including Exos secretion [13,15]. BMSC-Exos have been reported as therapeutic agents for the treatment of central nervous diseases due to the ability to cross the blood-brain barrier [28,29]. More important, exosomes not only show the same effects as BMSCs but also show the advantages of targeted delivery, low immunogenicity, and high biocompatibility. Cross-species administration of MSC-EVs was used in a variety of in vivo experimental models, in which the majority demonstrated beneficial outcomes to reflect the immunocompatibility of EVs [30]. In this study, we first identified the characteristics of BMSCs and BMSC-Exos and further found that BMSC-Exos improved neurological function and reduced infarct volume after cerebral ischemia in mice. Ni et al. also reported that rat BMSC-Exos exerted a neuroprotective function in traumatic brain injury mice [31]. Thus, our research presents new evidence that BMSC-Exos possess cross-species therapeutic effects following cerebral ischemia.

Nowadays, accumulating evidence has shown that angiogenesis is essential in brain tissue repair following stroke, and the promotion of angiogenesis is broadly recognized as a promising therapeutic strategy [32]. Angiogenesis is commonly defined as a multi-step process, involving proliferation, sprouting, migration, and tube formation from pre-existing vasculature [33]. In the present study, the results showed that BMSC-Exos could improve angiogenesis in ischemic stroke mice and promote the proliferation, migration, and tube formation of HUVECs in vitro. Angiogenesis is regulated by multiple proangiogenic factors, among which VEGF, VEGFR2, Ang-1, and Tie-2 play the most important roles. VEGF binds to its receptor VEGFR2 and triggers a downstream angiogenic signaling pathway, which promotes endothelial cell proliferation and migration and ultimately forms a new vascular tube [34,35]. However, the newly formed endothelial cell tubes are unstable due to the lack of pericytes and the formation of the perivascular extracellular matrix. The Ang-1/Tie-2 interaction regulates the maturation of newly formed vasculature, which eventually results in complex vascular network formation [36]. In this study, we found that BMSC-Exos could dramatically improve protein expressions of VEGF, VEGFR2, Ang-1, and Tie-2 in the ischemic boundary zone of stroke in mice. Double immunofluorescence results showed that these molecules were expressed in brain endothelial cells. However, we could only find the improved expression of VEGF and VEGFR2 in HUVECs after BMSC-Exos administration.

Exosomes can alter gene expression and bioactivity of recipient cells through transferring miRs [17]. It has been reported that MSCs promote angiogenesis via secreting exosomes that deliver pro-angiogenic miRs [37]. MiR-22 could regulate endothelial angiogenesis, inflammation, and tissue injury by targeting vascular endothelial-cadherin [38]. MiR-486 released by adipose-derived stem cell-derived EV could mediate wound healing and promote angiogenesis [39]. MiR-21-5p could promote angiogenesis in the unilateral anterior crossbite model and MCAO model [40,41]. More importantly, the miR expression profile of BMSC-Exos confirmed that let-7i-5p, miR-21-5p, miR-22-3p, and miR-486 were abundant in BMSC-Exos [21,42]. Thus, we detected the expression of let-7i-5p, miR-21-5p, miR-22-3p, and miR-486 in the ischemic boundary region of stroke in mice and HUVECs. qRT-PCR data indicated that BMSC-Exos upregulated the expression of these miRs, among which miR-21-5p showed the greatest effect. Then, the miR-21-5p inhibitor was transfected into BMSCs, and qRT-PCR results showed that the expression level of miR-21-5p was significantly decreased in BMSCs and BMSC-Exos. We also found that the expression level of miR-21-5p was significantly decreased in HUVECs treated with Exos derived from BMSCs transfected with miR-21-5p inhibitor. The proangiogenic effects of BMSC-Exos on HUVECs were greatly abolished by miR-21-5p inhibitor. Moreover, the expression levels of VEGF and VEGFR2 in HUVECs were also significantly reduced by Exos derived from BMSCs transfected with miR-21-5p inhibitor. These results suggest that the therapeutic effects of BMSC-Exos on angiogenesis of HUVECs may depend on miR-21-5p transportation.

This study also has some limitations. First, we did not detect miR expression profile of BMSC-Exos, nor the difference between BMSC-Exos and HUVECs. Although BMSC-derived exosomal miR-21-5p was demonstrated to promote angiogenesis in HUVECs in vitro, we could not rule out the possibility that BMSC-Exos increased endogenous miR-21-5p expression. Moreover, we also did not validate the proangiogenic activity of exosomal miR-21-5p by loss-of-function experiments in vivo, which could not determine whether BMSC-Exos promoted angiogenesis by transferring miR-21-5p or upregulating endogenous miR-21-5p in ischemic stroke in mice. Second, the target genes of miR-21-5p need to be further investigated. TargetScan, miRWalk, miRDB, and PicTar databases were used to predict the target genes of miR-21-5p. The results showed that reversion-inducing cysteine-rich protein with kazal motifs (RECK) was one of the target genes of miR-21-5p (data not shown). Some studies have reported that RECK plays an essential role in brain angiogenesis [43,44]. Therefore, it is necessary to confirm whether miR-21-5p plays a pro-angiogenic role in ischemic stroke by targeting RECK in the future.

## 5. Conclusions

In conclusion, our study indicates that BMSC-Exos can promote angiogenesis by upregulating miR-21-5p after stroke in mice, which provides new insights into the mechanism of BMSC-Exos that might be used in the treatment of ischemic stroke.

## Figures and Tables

**Figure 1 biomolecules-12-00883-f001:**
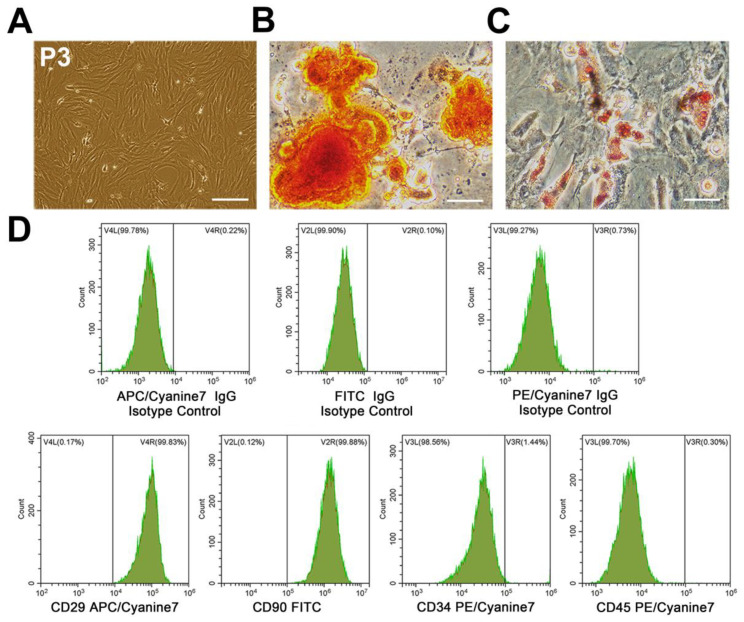
Characterization of BMSCs. (**A**) P3 BMSCs adopted a uniformly spindle-shaped population. Scale bars = 200 μm. (**B**) Osteogenic differentiation of BMSCs demonstrated by Alizarin Red S staining. Scale bars = 50 μm. (**C**) Adipogenesis differentiation of BMSCs demonstrated by Oil Red O staining. Scale bars = 50 μm. (**D**) Flow cytometric analysis of cell surface markers indicated that BMSCs expressed CD29 and CD90 other than CD34 and CD45. BMSCs, bone marrow mesenchymal stem cells; P3, passage 3.

**Figure 2 biomolecules-12-00883-f002:**
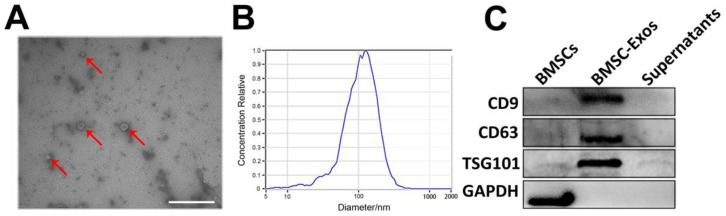
Identification of BMSC-Exos. (**A**) Cup-shaped morphology of purified BMSC-Exos (indicated with arrows) observed by TEM. Scale bar = 500 nm. (**B**) Particle size distribution and concentration of BMSC-Exos analyzed by NTA. (**C**) Western blots of exosomal membrane markers CD9, CD63, and TSG101. BMSC-Exos, bone marrow mesenchymal stem cell-derived exosome; TEM, transmission electron microscope; NTA, nanoparticle tracking analysis.

**Figure 3 biomolecules-12-00883-f003:**
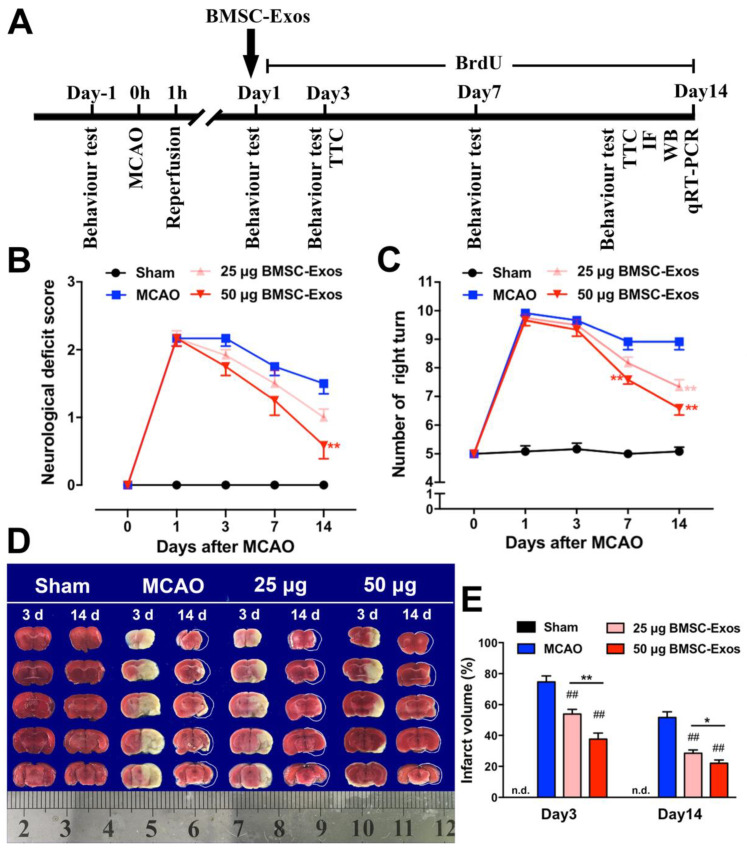
BMSC-Exos administration improved behavioral outcomes and reduced infarction volume in MCAO mice. (**A**) Schematic represents the timeline of in vivo experiments. (**B**) Neurological deficit scores. (**C**) Corner test. *n* = 12/group. ** *p* < 0.01 compared with MCAO. (**D**) Representative TTC stained images of mice coronal brain sections at 3 and 14 days after MCAO. White circumscribed line indicates pre-atrophy size of ischemic brain according to the size of contralateral brain. (**E**) Quantification of infract volume. Ruler at the bottom indicates size in centimeters. *n* = 12/group. ## *p* < 0.01 compared with MCAO. * *p* < 0.05 compared with 25 μg BMSC-Exos group. ** *p* < 0.01 compared with 25 μg BMSC-Exos group. BMSC-Exos, bone marrow mesenchymal stem cell-derived exosomes; MCAO, middle cerebral artery occlusion; TTC, 2,3,5-triphenyltetrazolium chloride; n.d., not detected.

**Figure 4 biomolecules-12-00883-f004:**
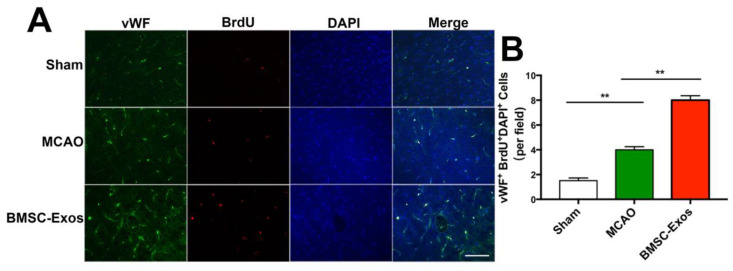
BMSC-Exos promoted angiogenesis in ischemic boundary zone. (**A**) Representative immunofluorescence images of the infarct boundary zone stained for vWF (green), BrdU (red) and DAPI (blue). Scale bar = 200 μm. (**B**) Quantification of vWF^+^/BrdU^+^/DAPI^+^ microvascular endothelial cells. *n* = 6/group. ** *p* < 0.01. vWF, von Willebrand factor; BrdU, 5-bromo-2-deoxyuridine; DAPI, 4′,6-diamidino-2-phenylindole; BMSC-Exos, bone marrow mesenchymal stem cell-derived exosomes; MCAO, middle cerebral artery occlusion.

**Figure 5 biomolecules-12-00883-f005:**
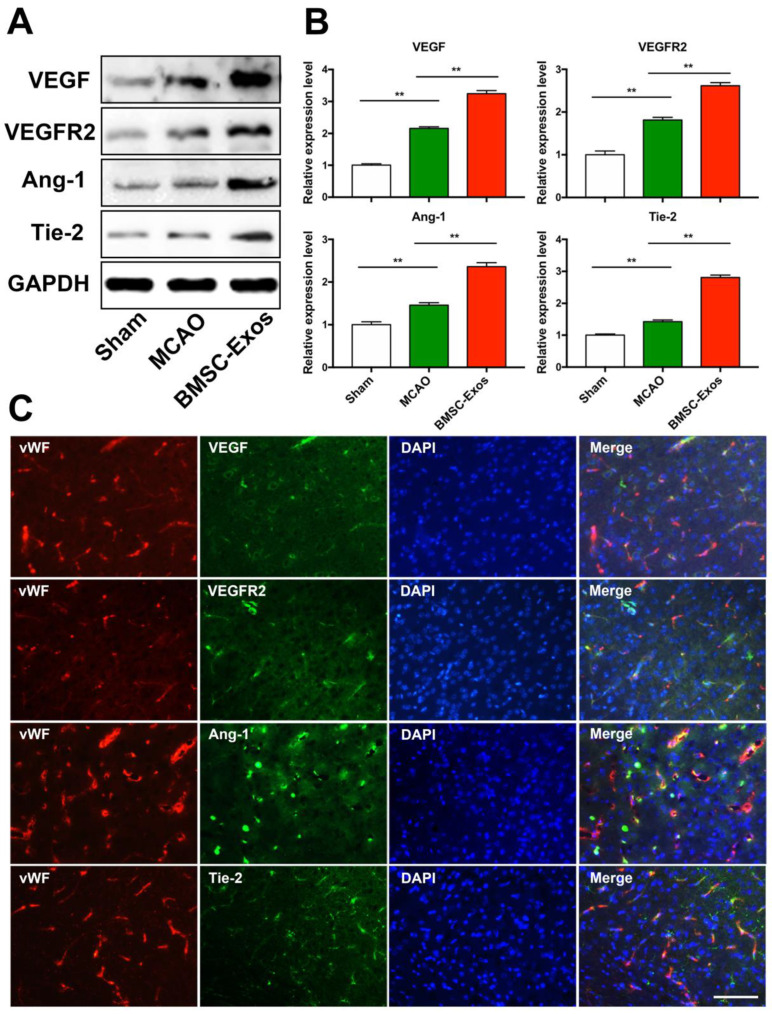
BMSC-Exos increased VEGF/VEGFR2 and Ang-1/Tie-2 protein expression after MCAO in mice. (**A**,**B**) Representative western blot analysis and quantification of densitometries of Western blot band. *n* = 3/group. ** *p* < 0.01. (**C**) Representative immunofluorescence images of the infarct boundary zone stained for vWF (red), VEGF/VEGFR2/Ang-1/Tie-2 (green), and DAPI (blue). Scale bar = 200 μm. vWF, von Willebrand factor; BrdU, 5-bromo-2-deoxyuridine; DAPI, 4′,6-diamidino-2-phenylindole; BMSC-Exos, bone marrow mesenchymal stem cell-derived exosomes; MCAO, middle cerebral artery occlusion; VEGF, vascular endothelial growth factor; VEGFR2, vascular endothelial growth factor receptor 2; Ang-1, angiogenin-1; Tie-2, tyrosine kinase receptor-2; GAPDH, glyceraldehyde-3-phosphate dehydrogenase.

**Figure 6 biomolecules-12-00883-f006:**
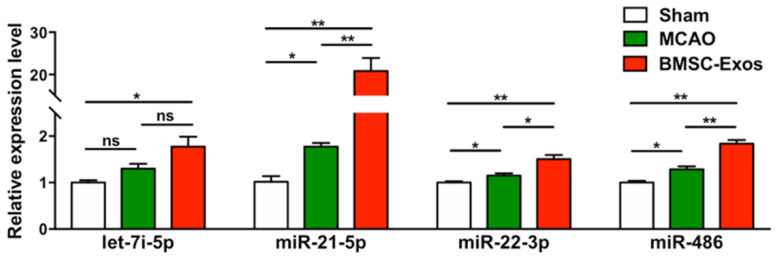
The expression levels of miRs in ischemic brains were determined by qRT-PCR. *n* = 3/group. * *p* < 0.05. ** *p* < 0.01, ns, no significance; BMSC-Exos, bone marrow mesenchymal stem cell-derived exosomes; MCAO, middle cerebral artery occlusion.

**Figure 7 biomolecules-12-00883-f007:**
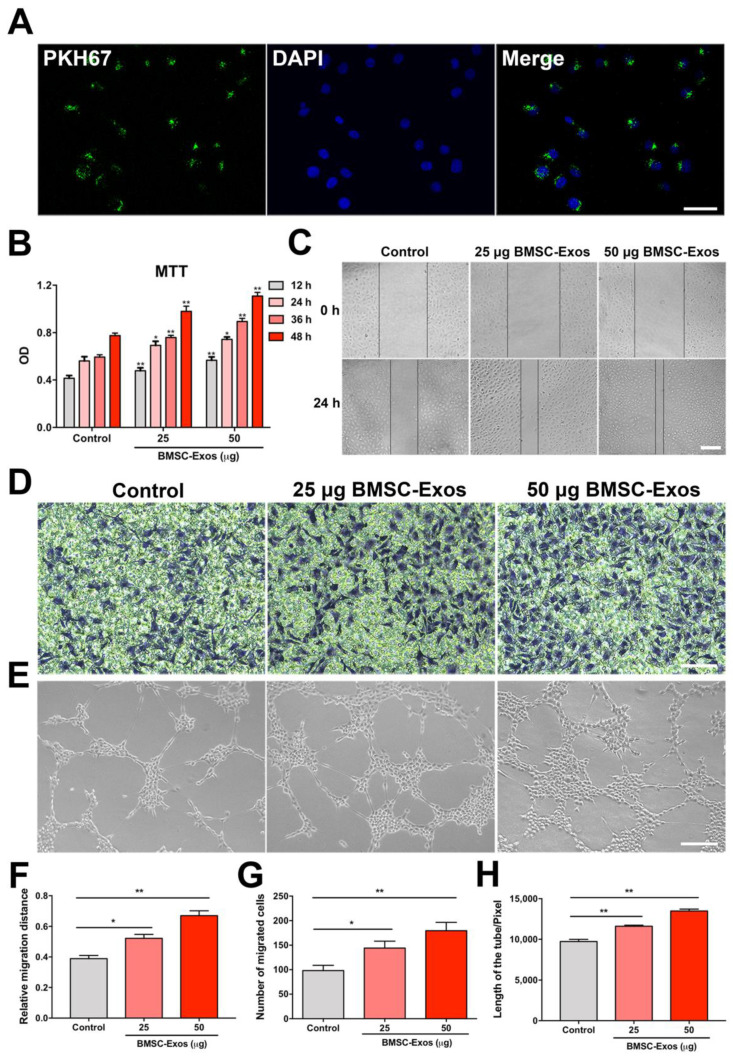
BMSC-Exos promoted proliferation, migration and tube formation of HUVECs. (**A**) PKH-labeled BMSC-Exos were uptake by HUVECs. Scale bar = 50 μm. (**B**) Cell viability was determined by MTT assay. *n* = 6/group. * *p* < 0.05. ** *p* < 0.01. (**C**) Representative images of scratch assay in HUVECs. Scale bar = 200 μm. (**D**) Representative images of migrated HUVECs in Transwell assay. Scale bar = 100 μm. (**E**) Representative images of tube formation of HUVECs. Scale bar = 200 μm. (**F**) Relative measurement of migration distance. *n* = 3/group. * *p* < 0.05. ** *p* < 0.01. (**G**) Number of migration cells in Transwell assay. *n* = 3/group. * *p* < 0.05. ** *p* < 0.01. (**H**) Total length of tube formed by HUVECs. *n* = 3/group. ** *p* < 0.01. DAPI, 4′,6-diamidino-2-phenylindole; MTT, 3-(4,5-dimethyl-2-thiazolyl)-2,5-diphenyl-2-H-tetrazolium bromide; OD, optical density; BMSC-Exos, bone marrow mesenchymal stem cell-derived exosomes.

**Figure 8 biomolecules-12-00883-f008:**
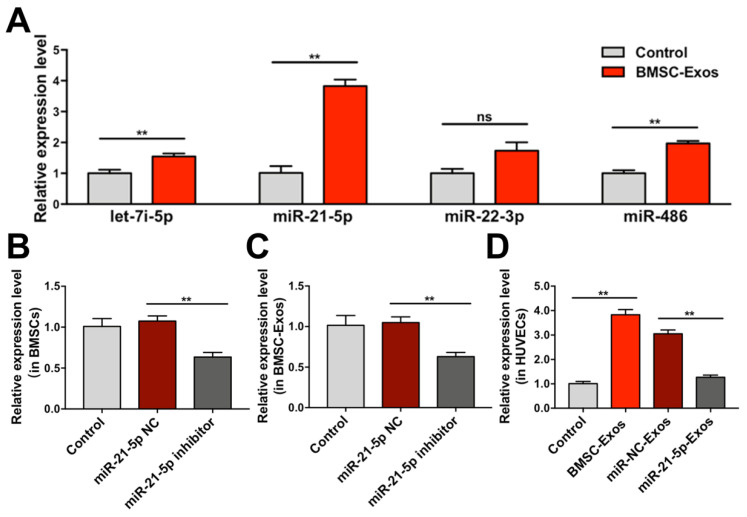
The expression of miRs in vitro were determined by qRT-PCR. (**A**) The expression levels of let-7i-5p, miR-21-5p, miR-22-3p and miR-486 in HUVECs treated with BMSC-Exos. ** *p* < 0.01. ns, not significant. (**B**) The expression levels of miR-21-5p in BMSCs after miR-21-5p inhibitor transfection. ** *p* < 0.01. (**C**) The expression levels of miR-21-5p in Exos derived from miR-21-5p inhibitor transfected BMSCs. ** *p* < 0.01. (**D**) The expression levels of miR-21-5p in HUVECs treated with Exos derived from miR-21-5p inhibitor transfected BMSCs. ** *p* < 0.01. *n* = 3/group. BMSCs, bone marrow mesenchymal stem cells; BMSC-Exos, bone marrow mesenchymal stem cell-derived exosome; miR-21-5p NC, miR-21-5p inhibitor negative control; miR-NC-Exos, Exos derived from BMSCs transfected with miR-21-5p inhibitor negative control; miR-21-5p-Exos, Exos derived from BMSCs transfected with miR-21-5p inhibitor.

**Figure 9 biomolecules-12-00883-f009:**
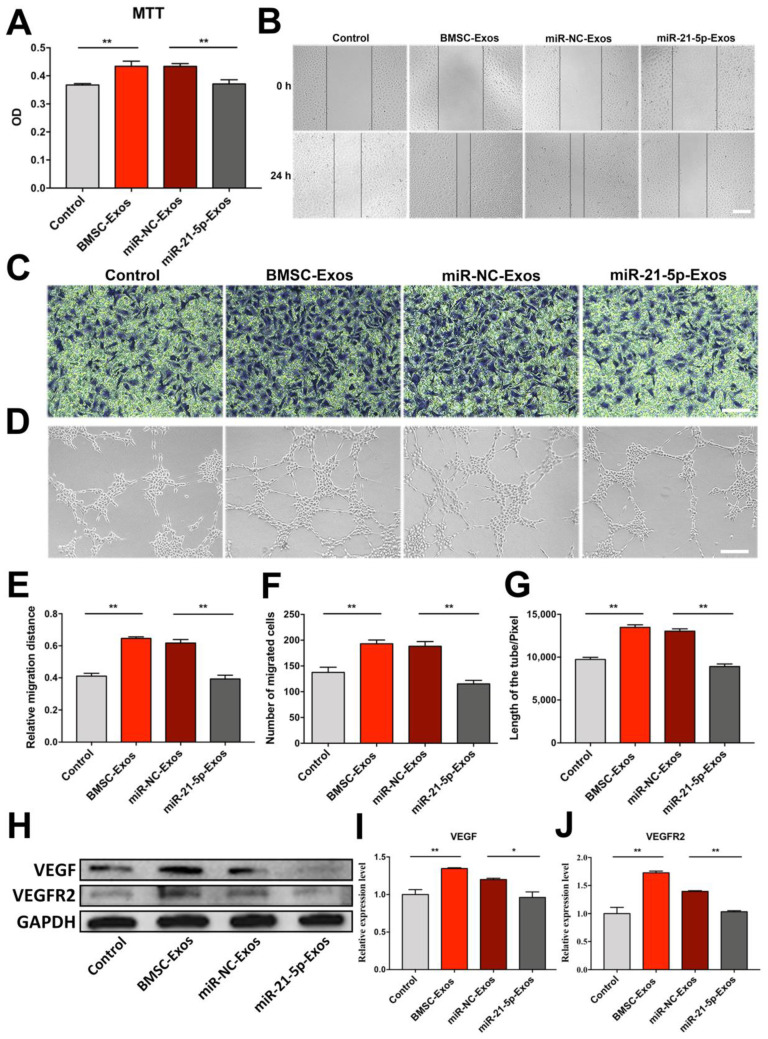
BMSC-Exos promoted proliferation, migration, and tube formation of HUVECs via upregulation of miR-21-5p. (**A**) Cell viability of HUVECs cocultured with miR-21-5p- or miR-NC-Exos. *n* = 6/group. ** *p* < 0.01. (**B**) Representative images of scratch assay for migration of HUVECs. Scale bar = 200 μm. (**C**) Representative images of Transwell assay for HUVECs migration. Scale bar = 100 μm. (**D**) Representative images of tube formation of HUVECs. Scale bar = 200 μm. (**E**) Measurement of relative migration distance. *n* = 3/group. ** *p* < 0.01. (**F**) Number of migration cells in Transwell assay. *n* = 3/group. ** *p* < 0.01. (**G**) Total length of tube formed by HUVECs. *n* = 3/group. ** *p* < 0.01. (**H**–**J**) Western blot analysis and quantification of densitometries of Western blot band. *n* = 3/group. * *p* < 0.05. ** *p* < 0.01. MTT, 3-(4,5-dimethyl-2-thiazolyl)-2,5-diphenyl-2-H-tetrazolium bromide; OD, optical density; BMSC-Exos, bone marrow mesenchymal stem cell-derived exosomes; miR-NC-Exos, Exos derived from BMSCs transfected with miR-21-5p inhibitor negative control; miR-21-5p-Exos, Exos derived from BMSCs transfected with miR-21-5p inhibitor; VEGF, vascular endothelial growth factor; VEGFR2, vascular endothelial growth factor receptor 2; GAPDH, glyceraldehyde-3-phosphate dehydrogenase.

**Table 1 biomolecules-12-00883-t001:** Primer sequences for qRT-PCR.

Primer	Sequence
miR-21-5p	5′-CCGCGTAGCTTATCAGACTCAGACTGATGTTGA-3′
miR-22-3p	5′-CGAAGCTGCCAGTTGAAGAACTGT-3′
miR-486	5′-TCCTGTACTGAGCTGCCCC-3′
let-7i-5	5′-GCGTGAGGTAGTAGTTTGTGCTGTT-3′

Note: The forward and reverse primers of U6, and mRQ 3′Primer used as universal reverse primer for miRs above were supplied in Mir-X miRNA First-strand Synthesis kit (Cat No. 638313).

## Data Availability

All data generated or analyzed during this study are available from the corresponding author upon reasonable request.

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
