# Peer review of "Exosomes Derived from Bone Marrow Mesenchymal Stem Cells Promote Angiogenesis in Ischemic Stroke Mice via Upregulation of MiR-21-5p"

_biomolecules, 2022, doi:10.3390/biom12070883_

Round 1

Reviewer 1 Report

The study presented by Hu et al. has investigated “Exosomes derived from bone marrow mesenchymal stem cells promote angiogenesis in ischemic stroke mice via upregulation of MiR-21-5p”. They studied whether BMSC-Exos could promote angiogenesis in ischemic stroke mice via miR-21-5p. The authors concluded that BMSC-Exos can promote angiogenesis by upregulation of miR-21-5p after stroke in mice. The results of this study are interesting. However, I have a few major concerns about the results of this study.

Majors

The Discussion doesn’t reflect the Results which should be discussed appropriately?

Graphs and pictographs should be improved for clarity, they are too small and hard to see.

It is not clear why the authors used Sprague-Dawley rats in their study while the experimental stroke was done in mice. The authors should provide the rational for their choice. The authors should also provide the number of animals used in each procedure.

It is not clear to me why the authors chose to present the data as mean ± SD instead of mean ± SEM? It is also not clear why they used one-way ANOVA?

Minors

Could the authors provide their gating strategy for the flow cytometry?

What was the mortality rate after MCAO and failing surgery in this study?

The title of section 3.4 should be changed to add “after stroke or MCAO” instead of after instead of after ischemia.

Is there any increased in VEGF/VEGFR2 and Ang-1/Tie-2 protein expression in the ischemic core?

Manuscript need proofread.

Author Response

Response to Reviewer 1 Comments

Majors

Point 1: The Discussion doesn’t reflect the Results which should be discussed appropriately?

Response 1: We are grateful for the suggestion. We have revised the Discussion according to Reviewer’s comment.

Point 2: Graphs and pictographs should be improved for clarity, they are too small and hard to see.

Response 2: We have made larger and clearer graphs according to Reviewer’s suggestion.

Point 3: It is not clear why the authors used Sprague-Dawley rats in their study while the experimental stroke was done in mice. The authors should provide the rational for their choice. The authors should also provide the number of animals used in each procedure.

Response 3: This comment is very important and helpful. Exosomes have recently emerged as a promising drug delivery system with low immunogenicity, high biocompatibility, and high efficacy of delivery. Furthermore, MSC-derived exosomes are viewed as potential mediators for induction of peripheral tolerance and modulation of immune responses [1, 2]. In 2020, Tieu et al used systematic review methodology to consolidate information from all published animal studies investigating mesenchymal stem cell-derived extracellular vesicles (MSC-EVs) as an intervention [3]. The results showed that species source of MSCs varied as well and included human (62%), mouse (20%), rat (16%), pig (2%), rabbit (1%) and dog (0.5%), and xenogeneic administration of EVs accounted for about 66% of all animals, a majority of which demonstrated beneficial outcomes to reflect the immunocompatibility of EVs [3]. Similarly, MSCs from one species can promote tissue recovery when transplanted into another species, resulting in improved function [4]. For example, human MSCs functioned in no fewer than 7 different recipient species [4].

In this study, to further confirm that exosomes transferred among different species, exosomes derived from rat BMSCs were used to treat ischemic stroke in mice. The results showed that treatment with exosomes derived from rat BMSCs ameliorated cerebral ischemia injury in mice. Ni et al reported that rat BMSC-Exos exerted a neuroprotective function by inhibiting neuroinflammation in traumatic brain injury mice [5].

We have explained this in the discussion section.

References:

[1] Rani S, Ryan AE, Griffin MD, Ritter T. Mesenchymal Stem Cell-derived Extracellular Vesicles: Toward Cell-free Therapeutic Applications. Mol Ther. 2015;23(5):812-823.

[2] Mokarizadeh A, Delirezh N, Morshedi A, Mosayebi G, Farshid AA, Mardani K. Microvesicles derived from mesenchymal stem cells: potent organelles for induction of tolerogenic signaling. Immunol Lett. 2012;147(1-2):47-54.

[3] Tieu A, Lalu MM, Slobodian M, et al. An Analysis of Mesenchymal Stem cell-Derived Extracellular Vesicles for Preclinical Use. ACS Nano, 2020;14(8):9728-9743. 

[4] Li J, Ezzelarab MB, Cooper DK. Do mesenchymal stem cells function across species barriers? Relevance for xenotransplantation. Xenotransplantation. 2012;19(5):273-85.

[5] Ni H, Yang S, Siaw-Debrah F, et al. Exosomes Derived From Bone Mesenchymal Stem Cells Ameliorate Early Inflammatory Responses Following Traumatic Brain Injury. Front Neurosci, 2019;13:14.

Point 4: It is not clear to me why the authors chose to present the data as mean ± SD instead of mean ± SEM? It is also not clear why they used one-way ANOVA?

Response 4: Thanks to the reviewer for helpful comments. The standard deviation (SD) measures the amount of variability or dispersion for a subject set of data from the mean, while the standard error of the mean (SEM) measures how far the sample mean of the data is likely to be from the true population mean. The SEM is more useful when reporting statistical results because it allows an intuitive comparison between the estimated populations via graphs or tables. With this in mind, we have revised all the statistical charts in the article according to your requirements. Please turn to the revised manuscript for details.

In this study, neurological deficit and corner test data were not normally distributed, nonparametric Kruskal-Wallis H test was used to characterize the probability of the data. All other data were random, independent, normal distribution, and homogeneity of variance, so statistical analysis was performed using one-way ANOVA.

Minors

Point 1: Could the authors provide their gating strategy for the flow cytometry?

Response 1: In this study, FITC Mouse IgG, APC/Cyanine7 Armenian Hamster IgG and PE/Cyanine7 Mouse IgG1 Isotype Control were used to determine nonspecific signals and set gating strategy.

We also added the Isotype Control in Figure 1D.

Point 2: What was the mortality rate after MCAO and failing surgery in this study?

Response 2: In this study, the mortality rate after MCAO at 14 days was about 55.84%, and the failing surgery rate was 14.81%.

Point 3: The title of section 3.4 should be changed to add “after stroke or MCAO” instead of after instead of after ischemia.

Response 3: Thank you for pointing out our mistakes. As suggested by the reviewer, we have corrected the “after ischemia” into “after MACO”.

Point 4: Is there any increased in VEGF/VEGFR2 and Ang-1/Tie-2 protein expression in the ischemic core?

Response 4: VEGF/VEGFR2 and Ang-1/Tie-2 play important roles in the process of angiogenesis. Several studies have reported their temporal and spatial expression patterns after MCAO. In general, VEGF/VEGFR2 expression were increased in both ischemic core and penumbra during the early stage of cerebral ischemia and in the peri-infarct area during the late stage of cerebral ischemia [1-3]. Ang-1/Tie-2 were mainly expressed in the peri-infarct area [4-5]. Therefore, in our study, we observed VEGF/VEGFR2 and Ang-1/Tie-2 expression in the ischemic boundary zone at 14 days after MCAO. The results showed that the expression levels of these proteins were increased, which was consistent with previous studies [6-7].

References:

[1] Plate KH, Beck H, Danner S, et al. Cell type specific upregulation of vascular endothelial growth factor in an MCA occlusion model of cerebral infarct. J Neuropathol Exp Neurol. 1999;58:654-66.

[2] Marti HJ, Bernaudin M, Bellail A, et al. Hypoxia-induced vascular endothelial growth factor expression precedes neovascularization after cerebral ischemia. Am J Pathol. 2000;156:965-76.

[3] Kovacs Z, Ikezaki K, Samoto K, et al. VEGF and flt. Expression time kinetics in rat brain infarct. Stroke. 1996;27(10):1865-72; discussion 1872-3.

[4] Beck H, Acker T, Wiessner C, et al. Expression of angiopoietin-1, angiopoietin-2, and tie receptors after middle cerebral artery occlusion in the rat. Am J Pathol. 2000;157(5):1473-83.

[5] Zan LK, Song YJ, Teng GX, et al. Expression and function of vascular endothelial growth factor and angiopoietins in rat brain after cerebral ischemia. Zhonghua Bing Li Xue Za Zhi. 2011;40(12):834-9.

[6] Janyou A, Wicha P, Seechamnanturakit V, et al. Dihydrocapsaicin-induced angiogenesis and improved functional recovery after cerebral ischemia and reperfusion in a rat model. J Pharmacol Sci. 2020;143(1):9-16.

[7] He FY, Ma CJ, Feng J, et al. Angiogenesis effects of 4-methoxy benzyl alcohol on cerebral ischemia-reperfusion injury via regulation of VEGF-Ang/Tie2 balance. Can J Physiol Pharmacol. 2021;99(12):1253-63.

Point 5: Manuscript need proofread.

Response 5: We have invited Prof. Qinghua Sun from Ohio State University to check whole of the text for grammatical style and word use.

Special thanks to you for your good comments.

Reviewer 2 Report

A nice study which is novel and impactful. The manuscript is by and large well written and presented in a clear and concise fashion. The data is well presented, interrogated and interpreted. Materials and Methods are comprehensively described. Overall it is an informative and thought provoking study, in a translationally and clinically important field. I would foresee the manuscript to be of interest to a wide readership as it contributes new findings to the therapeutic use/effect of exosomes and their cargo (miRNA), IS in this instance. Minor English revision and editing required (syntax/grammar). I've listed a couple below.

Ln39; researches to studies (syntax)

Ln46-48; 'Nevertheless, following researches reported that the low engraftment and poor survival 46 of transplanted MSCs in the ischemic regions could not sufficiently explained the differentiation mechanisms [8]'. Please revise syntax.

Ln88; 'Then, centrifuged the bone marrow aspirates' - to bone marrow aspirates were then centrifuged. (Syntax).

Ln116; syntax- quantified by total proteins

Minor comments-

1. Can the authors give a background rational as to why the miRNA analysed were selected, to miR-21-5p in particular. Were these identified in a screen?

2. Was the miRomic profile of Bone Marrow Mesenchymal Stem Cell derived Exosomes profiled? The observed up regulation of miR-21-5p in HUVECs- did endogenous expression (in HUVECs) contribute anything to this, i.e. did treating the HUVECs with BMMSC-Exos up regulate HUVEC miR-21-5p expression? Was the miRomic profiles of HUVECs compared and contrasted to under various regimes in parallel to Exosome profiles? While this information is not essential for this paper, if the authors had background and supporting data that would be interesting. I think this would help define and explain the issue of 'causation or correlation' to the findings. If these studies are not at hand, the authors should address this in the manuscript with supporting references.

Author Response

Response to Reviewer 2 Comments

Point 1: A nice study which is novel and impactful. The manuscript is by and large well written and presented in a clear and concise fashion. The data is well presented, interrogated and interpreted. Materials and Methods are comprehensively described. Overall it is an informative and thought provoking study, in a translationally and clinically important field. I would foresee the manuscript to be of interest to a wide readership as it contributes new findings to the therapeutic use/effect of exosomes and their cargo (miRNA), IS in this instance. Minor English revision and editing required (syntax/grammar). I've listed a couple below.

Response 1: Thank you very much for revision of our manuscript and your positive feedback. We have invited Prof. Qinghua Sun from Ohio State University to check whole of the text for grammatical style and word use.

Point 2: Ln39; researches to studies (syntax)

Response 2: We have changed “researches” to “studies”.

Point 3: Ln46-48; 'Nevertheless, following researches reported that the low engraftment and poor survival 46 of transplanted MSCs in the ischemic regions could not sufficiently explained the differentiation mechanisms [8]'. Please revise syntax.

Response 3: We have rewritten this sentence according to reviewer’s comment.

Point 4: Ln88; 'Then, centrifuged the bone marrow aspirates' - to bone marrow aspirates were then centrifuged. (Syntax).

Response 4: We have rewritten this sentence according to reviewer’s comment.

Point 5: Ln116; syntax- quantified by total proteins

Response 5: We have rewritten this sentence according to reviewer’s comment.

Minor comments

Point 1: Can the authors give a background rational as to why the miRNA analysed were selected, to miR-21-5p in particular. Were these identified in a screen?

Response 1: These comments are really important and helpful. MiRNA regulate the expression of approximately 30% of human genes and play important regulatory roles in many diseases. Several studies have reported that BMSC-Exos contain diverse and numerous miRNAs, mainly exert their function by transferring exosomal miRNA to recipient cells. For example, growing evidence has shown that exosomes transfer miRs to the recipient cells in the brain to affect the nervous and vascular systems, and ultimately alleviate cerebral ischemic injury [1-5]. Previous studies have revealed that miR-21-5p, miR-22-3p, miR-486, and let-7i-5p were abundant in BMSC-Exos by small RNA-sequencing [6, 7]. Therefore, we detected the expression levels of these four miRNAs in the ischemic brain tissue and HUVECs after BMSC-Exos treatment, among which miR-21-5p showed the greatest effect. Then, we further demonstrated that the promoted HUVECs angiogenesis by BMSC-Exos was mediated through upregulated miR-21-5p by loss-of-function experiments in vitro. This work could suggest that the pro-angiogenic effect of BMSC-Exos was partly mediated by miR-21-5p.

We have provided a background rational in the Introduction section.

References:

[1] Yang JH, Gao F, Zhang YK, et al. Buyang Huanwu Decoction (BYHWD) Enhances Angiogenic Effect of Mesenchymal Stem Cell by Upregulating VEGF Expression After Focal Cerebral Ischemia.  J Mol Neurosci, 2015;56(4):898-906. 

[2] Zhang HX, Wu J, Wu JH, et al. Exosome-mediated targeted delivery of miR-210 for angiogenic therapy after cerebral ischemia in mice.  J Nanobiotechnology, 2019;17(1):29.

[3] Wang JJ, Chen SZ, Zhang WF, et al. Exosomes from miRNA-126-modified endothelial progenitor cells alleviate brain injury and promote functional recovery after stroke. CNS Neurosci Ther, 2020;26(12):1255-1265. 

[4] Hou K, Li G, Zhao J, et al. Bone mesenchymal stem cell-derived exosomal microRNA-29b-3p prevents hypoxic-ischemic injury in rat brain by activating the PTEN-mediated Akt signaling pathway. J Neuroinflammation, 2020;17(1):46.

[5] Yang Y, Cai Y, Zhang Y, et al. Exosomes Secreted by Adipose-Derived Stem Cells Contribute to Angiogenesis of Brain Microvascular Endothelial Cells Following Oxygen-Glucose Deprivation In Vitro Through MicroRNA-181b/TRPM7 Axis. J Mol Neurosci, 2018;65(1):74-83.

[6] Ferguson SW, Wang J, Lee CJ, et al. The microRNA regulatory landscape of MSC-derived exosomes: a systems view. Sci Rep, 2018;8(1):1419.

[7] Baglio SR, Rooijers K, Koppers-Lalic D, Verweij FJ, Perez Lanzon M, Zini N et al. Human bone marrow- and adipose-mesenchymal stem cells secrete exosomes enriched in distinctive miRNA and tRNA species. Stem Cell Res Ther, 2015;6:127.

Point 2: Was the miRomic profile of Bone Marrow Mesenchymal Stem Cell derived Exosomes profiled? The observed up regulation of miR-21-5p in HUVECs- did endogenous expression (in HUVECs) contribute anything to this, i.e. did treating the HUVECs with BMMSC-Exos up regulate HUVEC miR-21-5p expression? Was the miRomic profiles of HUVECs compared and contrasted to under various regimes in parallel to Exosome profiles? While this information is not essential for this paper, if the authors had background and supporting data that would be interesting. I think this would help define and explain the issue of 'causation or correlation' to the findings. If these studies are not at hand, the authors should address this in the manuscript with supporting references.

Response 2: Thanks for your constructive comments. We did not detect the miRomic profile of bone marrow mesenchymal stem cell derived exosomes profiled. In this study, we found that BMSC-Exos could up-regulate miR-21-5p expression in ischemic boundary region of stroke in mice and in HUVECs (Fig. 6 and 8A). Then, BMSCs were transfected with miR-21-5p inhibitor. The qRT-PCR showed that miR-21-5p expression was significantly downregulated in BMSCs and BMSC-Exos (Figure 8B, 8C). Treatment with Exos derived from BMSCs transfected with miR-21-5p inhibitor (miR-21-5p-Exos) also significantly decreased miR-21-5p expression in HUVECs (Fig. 8D). These results indicated that BMSC-Exos increased miR-21-5p expression in HUVECs by transferring miR-21-5p. However, we cannot rule out the possibility that BMSC-Exos increase endogenous miR-21-5p expression. Fig. 8C showed that treatment with Exos derived from BMSCs transfected with miR-21-5p inhibitor did not completely reverse miR-21-5p expression in HUVECs.

We did not compare and contrast whether the miRomic profiles of HUVECs in parallel to Exosome profiles under various regimes. But this suggestion is very important and valuable. We will compare the the miRomic profiles between HUVECs and exosomes in future study. We have addressed this in the manuscript with supporting references.

Special thanks to you for your good comments.

Round 2

Reviewer 1 Report

Professor Qinghua Sun from Ohio State University should be acknowledged for proofreading the manuscript.